# Ecology of Saline Watersheds: An Investigation of the Functional Communities and Drivers of Benthic Fauna in Typical Water Bodies of the Irtysh River Basin

**DOI:** 10.3390/biology13010027

**Published:** 2024-01-02

**Authors:** Fangze Zi, Baoqiang Wang, Liting Yang, Qiang Huo, Zhichao Wang, Daoquan Ren, Bin Huo, Yong Song, Shengao Chen

**Affiliations:** 1Tarim Research Center of Rare Fishes, College of Life Sciences and Technology, Tarim University, Alar 843300, China; 10757213076@stumail.taru.edu.cn (F.Z.); 10757231127@stumail.taru.edu.cn (L.Y.); 10757223082@stumail.taru.edu.cn (Q.H.); wzcdky@taru.edu.cn (Z.W.); 119950007@stumail.taru.edu.cn (D.R.); 2Institute of Hydrobiology, Chinese Academy of Sciences, Wuhan 430072, China; wangbq@ihb.ac.cn; 3College of Fisheries, Huazhong Agricultural University, Wuhan 430070, China; huobin@mail.hzau.edu.cn

**Keywords:** saline, subsidiary waters, functional traits, benthic fauna

## Abstract

**Simple Summary:**

In this study, we explored the impact of saline–alkaline environments on the diversity and functional traits of benthic organisms. In environments with varying levels of salt, we noticed changes in living organisms, like improved swimming abilities and better adaptation to breathing in high-salinity conditions. These findings offer crucial insights to understand and manage ecosystems in saline–alkaline environments, particularly in the face of climate change. This study provides a practical foundation for conservation efforts, emphasizing the importance of recognizing the unique roles of benthic organisms across varying salinity levels. By revealing adaptive strategies and their ecological significance, our research contributes to global initiatives that address environmental challenges and promote enhanced ecosystem resilience.

**Abstract:**

In this study, we investigated how changes in salinity affect biodiversity and function in 11 typical water bodies in the Altai region. The salinity of the freshwater bodies ranged from 0 to 5, the brackish water salinities ranged from 5 to 20, and the hypersaline environments had salinities > 20. We identified 11 orders, 34 families, and 55 genera in 3061 benthic samples and classified them into 10 traits and 32 categories. Subsequently, we conducted Mantel tests and canonical correlation analysis (CCA) and calculated biodiversity and functional diversity indices for each sampling site. The results indicated that biodiversity and the proportion of functional traits were greater in freshwater environments than in saline environments and decreased gradually with increasing salinity. Noticeable shifts in species distribution were observed in high-salinity environments and were accompanied by specific functional traits such as swimming ability, smaller body sizes, and air-breathing adaptations. The diversity indices revealed that the species were more evenly distributed in high-diversity environments under the influence of salinity. In contrast, in high-salinity environments, only a few species dominated. The results suggested that increasing salinity accelerated the evolution of benthic communities, leading to reduced species diversity and functional homogenization. We recommend enhancing the monitoring of saline water resources and implementing sustainable water resource management to mitigate the impact of salinity stress on aquatic communities in response to climate-induced soil and water salinization.

## 1. Introduction

Inland lakes are recognized as one of Earth’s most crucial aquatic ecosystems and are critical for human activities and production in many regions [1,2]. However, in recent years, most inland lakes worldwide have been affected by climate change factors such as global warming, carbon emissions, and seasonal precipitation [3,4,5]. Global climate change has led to compelling alterations in the aquatic environments of inland lakes, subsequently impacting aquatic biodiversity and community structure [6]. Although the driving factors affecting lake environments are generally similar, the mechanisms underlying the development of aquatic communities vary greatly [7,8]. As integral components of aquatic ecosystems, benthic fauna exhibit complex interactions between their adaptive strategies and environmental factors, critically influencing ecosystem stability and functionality [9]. The presence and behavior of benthic animals can be considered “indicators” of ecosystem health, as they are susceptible to environmental changes; thus, benthic animal behavior is crucial in ecological monitoring and conservation efforts [10,11].

As one of the most critical aquatic environmental factors, salinity is crucial for shaping the nutritional strategies, life history characteristics, and ecological niche distribution of benthic fauna [12,13]. However, with changes in salinity gradients, benthic organisms may adjust their feeding behaviors and ecological roles, impacting ecosystem stability and functionality [14]. For example, as salinity increases, benthic communities exhibit reduced diversity and increased homogenization of functional traits [15,16,17]. Different salinity levels create distinct environmental pressures on benthic organisms, leading to variations in their ecological strategies [18,19]. In environments characterized by low salinity, there is a proliferation of freshwater species, and, conversely, high-salinity conditions support an increased abundance of marine species. Notably, these organisms demonstrate a distinct minimum quantity at intermediate salinities, thereby shaping the foundational framework of our understanding of biodiversity along coastal salinity gradients. This pivotal concept is effectively visualized through Remane curves [20].

China’s unique high-altitude and cold climate conditions have created one of the most distinctive habitats for cold-water aquatic organisms in inland areas [4,21]. The Altai region features a diverse geographic environment, including mountains, gorges, plateaus, grasslands, and deserts [22]. Despite its rich water resources, including rivers, lakes, and wetlands, the Altai region has been increasingly affected by climate change and human activities, leading to the salinization of soils and water [23,24]. Due to their complex geographic environment and precipitation patterns, lakes in these regions exhibit significant variations in salinity [25,26]. Freshwater lakes are primarily situated in alpine valleys and rely on snowmelt for their water supply. Consequently, their salinity levels remain low, fostering the thriving development of cold-water aquatic organisms in alpine regions, as exemplified by Kanas Lake, among others [27]. Reservoirs, classified as artificial water bodies, are typically constructed at mountain outflows, particularly in Altai mountain passes [28]. The primary purpose of these plants is to collect and store snowmelt water, serving diverse functions, such as agricultural irrigation, water supply, and power generation. Nevertheless, human alterations in water resource distribution have impacted inland water environments [29,30]. In the middle and lower plains of rivers, elevated mineral salt content in water and soil combined with reduced water flow leads to gradual salt accumulation, resulting in high-salinity conditions. This phenomenon is notable in salt lakes, which are predominantly found in arid plains, as exemplified by the findings of Alahak Salt Lake. Consequently, the utilization of inland water resources and soil salinization pose significant environmental challenges.

Through field investigations, sample collections, and detailed data analysis, we aimed to comprehensively explore the ecological attributes of benthic fauna in diverse salinity environments, encompassing aspects such as diversity, functional characteristics, and ecological niche differentiation. We conducted a thorough examination of the impact of salinity gradients on the distribution of functional characteristics among typical benthic fauna in the water bodies of the Altay region. Our objective was to offer scientific theoretical support and practical guidance for the sustainable management and conservation of aquatic ecosystems in this region. Furthermore, our commitment extends to enhancing our understanding of the ecological adaptations and functions of benthic fauna under varying salinity conditions, contributing to a profound comprehension of their ecological roles within aquatic ecosystems.

## 2. Materials and Methods

### 2.1. Field Sampling and Data Acquisition

Considering geographical and environmental attributes, we collected benthic samples in June 2023 from 17 representative water bodies of interest (Figure 1). During each sampling event, we conducted triplicate measurements of water temperature (WT, °C), dissolved oxygen (DO, mg/L), salinity (SAL, ‰), pH, and oxidation-reduction potential (ORP, mV) using a multiparameter water quality sampler (YSI 556MPS, Xylem Analytics, OH, USA). Conforming to the criteria outlined in the Analytical Methods for Water and Wastewater Monitoring (4th edition), we employed a 1 L plexiglass water collector for procuring water quality samples. Additionally, water samples were acquired to assess total phosphorus (TP, mg/L), total nitrogen (TN, mg/L), chloride ions (Cl^−^), nitrate (NO_3_^−^), and ammonia ions (NH_4_^+^).

We employed the multihabitat sampling approach; an area of 0.3 m^2^ of benthic fauna was procured using a D-shaped net (0.3 m wide, 500 μm diameter) traversing the 100 m lakeshore stretch. The allocation of sampling squares was contingent upon the occurrence distribution ratios of distinct habitat categories (e.g., pools, shallows, sandy areas). The three sampled squares from each sampling location were amalgamated to constitute a single site sample, which was subjected to filtration via a 60-mesh sieve and stabilized within a solution of analytically pure ethanol. All benthic individuals were meticulously sorted, identified, and tallied in the laboratory. The taxonomic assignment of all entities was carried out to the nearest taxonomic level based on credible information within the laboratory. As a rule, taxa such as Ephemeroptera, Trichoptera, and Odonata were classified at the genus level. Furthermore, restricted portions of Coleoptera, Hemiptera, and Diptera were used to categorize the family ranks. We refer to Michael T. Barbour (Rapid bioassessment protocols for use in streams and wadeable rivers: Periphyton, benthic macroinvertebrates, and fish; 2nd edition). A selection encompassing ten traits that influenced ecosystem functionality, encompassing aspects such as morphology (adult size, extent of exoskeleton coverage) and behavior (perching mode, feeding habits), cumulatively totaling 32 distinct trait categories (Table 1), was used to scrutinize the functional attributes of macrobenthos. Biological trait data were acquired from published literature and online databases [31,32]. Fuzzy codes (ranging from 0 to 3) were used to allocate values to the trait types. A value of 0 signified the incompatibility of the trait type with the species, whereas 3 denoted a pronounced degree of compatibility. The codes were subsequently standardized post assignment. Certain species exhibited various behavioral traits, necessitating the assignment of multiple trait categories to a single biological trait. For instance, if certain species exhibited dual feeding modes, with both modes accounting for identical proportions, both feeding modes of said species could be attributed to a value of 2. Functional traits were allocated predominantly at the species level whenever feasible. When direct references for the functional trait were lacking, the assignment was executed by referencing the functional trait value at the genus or family level.

### 2.2. Data Analysis

#### 2.2.1. Correlation Analysis between Functional Traits and Environmental Factors of Benthic Organisms

The Mantel test and canonical correlation analysis (CCA) were used to explore the correlations and interrelationships between the functional traits of benthic animals and environmental factors.

Mantel test: Initially, a Mantel test was conducted to evaluate the correlation between benthic functional traits and environmental factors. Pearson’s correlation coefficients were computed for the matrices of functional traits and environmental factors. Correlation coefficients were also computed between the geographical distance matrix and the functional trait matrix to mitigate the impact of geographical distance. Through 999 rounds of data randomization, the significance level of the original correlation coefficients was computed.

Canonical correlation analysis (CCA): Building upon the Mantel test results, we used CCA to investigate the intricate associations between benthic functional traits and environmental factors. As mentioned earlier, we selected the functional traits and environmental factors that exhibited significant correlations in the Mantel test to serve as inputs for the CCA. During the CCA, the data underwent standardization to mitigate the impact of diverse scales. Subsequently, a CCA model was formulated, with functional traits as response variables and environmental factors as explanatory variables. The Monte Carlo permutation test was employed to evaluate the statistical significance of the CCA model. Additionally, we computed the redundancy values of the environmental factors along the CCA1 axis, thereby ascertaining their explanatory influence on the functional traits.

#### 2.2.2. Biodiversity

For species diversity indices, we used the Pielou (*J*) diversity index to analyze the data and results, which were calculated using the following formulas:(1)$$H^{\prime }=-\sum \left(N_{i}/N\right)log_{2}\left(N_{i}/N\right)$$
(2)$$J=H^{\prime }/log_{2}S\;$$
where *N_i_* is the number of individuals of species i, *N* is the total number of individuals of all species, and *S* is the number of species.

#### 2.2.3. Functional Diversity

We utilized functional richness, functional evenness, and functional divergence to assess functional diversity. FRic is a functional volume index that calculates the volume of a trait space using the volume of a minimum convex polygon, depicting the minimum convex polygon that contains all traits; i.e., there must be a minimum convex polygon in the trait space such that the points of all species fall within its range or on its edges. First, we identified the species with the most extreme values of the trait as the endpoints of the minimum convex polygon; subsequently, we connected them to generate the minimum convex polygon and, finally, calculated the polygon area or volume. FEve represents the degree to which the functional traits of a single individual are evenly distributed across the ecological space. It is proportional to the degree of adequate resource utilization within the ecological space. FDiv refers to the difference in the functional traits of individuals within the biological community and is directly proportional to the degree of complementarity of the ecological niche and inversely proportional to the degree of resource competition. FDis indicates the degree of functional discretization between species in the community, and a high index indicates a high degree of positional differentiation in ecological niches. The distance variation between species was determined using Rao’Q analysis. The macroinvertebrate functional trait values were calculated from the community-weighted average trait index with the following equation:(3)$$FRic=SF_{ce}∕R_{c}$$
(4)$$FEve=\frac{\sum_{I=1}^{S-1}min(PEW_{i}\times \frac{1}{S-1})-\frac{1}{S-1}}{1-\frac{1}{S-1}}$$
(5)$$FDiv=\frac{\Delta d+\frac{1}{S}\sum_{i=1}^{s}dG_{i}}{\Delta \left|d\right|+\frac{1}{S}\sum_{i=1}^{s}dG_{i}}$$
(6)$$FDis=\sum_{i=1}^{S}\min(P_{i},\frac{1}{S})$$
(7)$$\mathrm{Rao}\prime Q=\sum_{i=1}^{s}\sum_{j=1}^{s}d_{ij}P_{i}P_{j}$$
where *SFce* is the ecological niche space occupied by trait c within community e, *R_c_* is the absolute value of trait c, *S* is species richness, *PEWi* is the weighted evenness of species i, *dGi* is the mean distance of the Euclidean distances from species *S* to the center of gravity, *Δd* is the sum of abundance-weighted biases, Δ|*d*| is the sum of absolute weighted biases, and *d_ij_* is the degree of difference in the functional traits between species i and species j. *P_i_* and *P_j_* are the relative plurality of species i and j.

All analyses were performed in “R” version 3.5.1 (AT&T BellLaboratories, Auckland, New Zealand). We used the following packages: The “linkET” package for the Mantel test; the “FD” package for the functional diversity index; and the “vegan”, “ggrepel”, “ggplot2”, and “ggpubr” packages for the CCA.

## 3. Results

### 3.1. Characterization and Distribution

In our study, 3061 benthic samples were collected, covering 55 genera from 11 orders and 34 families through taxonomic identification. Among this diverse array of species, Insecta species dominated, constituting 83% of the total species. These species were followed by Hirudinea (7%), Oligochaeta (4%), Crustacea, and Gastropoda, each contributing 3% to the total species count. In freshwater environments, benthic organisms from ten different orders were identified. The Basommatophora, Amphipoda, and Rhynchobdellida orders were exclusively found in freshwater habitats (Figure 2). When considering relative abundance, the Diptera and Hemiptera orders dominated in freshwater habitats. Seven different orders of benthic organisms were identified in saline environments, with Decapoda exclusively present in saline waters. However, the abundances of the orders Trichoptera and Odonata significantly increased in saline environments, surpassing their presence in freshwater habitats and crucially influencing the composition of saline benthic communities. In high-salinity environments, the species distribution sharply decreased, with only four orders identified, and their relative abundance was notably lower than that of their counterparts in the other two environments.

In our study, we extensively analyzed the functional trait distributions in benthic communities under different salinity conditions (Figure 3). A clear trend emerged across the three salinity groups. In freshwater environments, most traits exceeded 50%, highlighting their prevalence in low-salinity habitats. However, in higher-salinity environments, the proportion of each trait gradually decreased, with a notable 22% reduction. Lightly covered exoskeleton traits stood out in saline environments, showing a significantly higher proportion than other traits and distinct separation in cluster analysis. The widespread presence of these organisms in saline environments underscores their crucial adaptive role in survival.

Under high-salinity conditions, the benthic organisms in our study exhibited functional traits related to swimming, small body size (<9 mm), and adaptations for air breathing (respiratory pores, tracheae, and dorsal armor) (Figure 3). These traits collectively represented essential strategies for coping with the challenges of high-salinity environments. In contrast, certain traits, such as gatherers–predators, climbers, adherents, multiple breeders, and swimming ability, made up a relatively small proportion of high-salinity environments. This contrast highlights the specialized adaptations necessary for survival under varying salinity conditions. Our analysis, which involved enrichment through cluster analysis, revealed strong associations between several functional trait categories. For example, small body size (<9 mm), terrestrial locomotion, and poor swimming ability were significantly correlated, indicating their common coexistence and ecological interrelatedness. Similarly, cluster analysis revealed associations between the integument, nonstreamlined morphology, soft body shape, and gatherer–filterers, suggesting that there are shared ecological strategies among these trait categories.

### 3.2. Environmental Factors and Functional Traits

Our results revealed significant correlations between environmental factors and benthic communities in environments with different salinities, providing valuable insights into the complex interactions between variables within aquatic ecosystems. Specifically, our research revealed a robust positive correlation between salinity (SAL) and total dissolved solids (TDS) concentration. A maximum correlation coefficient of 1 was observed in freshwater and high-salinity environments, highlighting the robust and consistent relationship between salinity and total dissolved solids in these habitats. Furthermore, significant associations involving dissolved oxygen (DO) were found in freshwater environments, where DO exhibited positive correlations with TDS, SAL, and pH. The correlation coefficients for these relationships were 0.57, 0.58, and 0.61, emphasizing the intricate interplay between these vital environmental factors (Figure 4a).

We also explored interactions related to nutrient dynamics involving ammonia (NH_4_^+^) and its associations with chloride (Cl^−^) and nitrate (NO_3_^−^). The correlation coefficients for these interactions were −0.95 and −0.62, indicating significant negative relationships. Importantly, these patterns were similar to what was observed in saline environments, reaffirming the ecological importance of these relationships. Additionally, we investigated functional traits to elucidate the complex relationships between biological features and various environmental factors. Notably, in freshwater environments, NH_4_^+^ concentrations exhibited significant positive correlations with functional traits, including exoskeletal or external protection, habitat, morphology, and feeding habits (Figure 4b). These correlations were consistent with the environmental conditions in freshwater habitats.

Conversely, in high-salinity environments, NH_4_^+^ concentrations were negatively correlated with body size, respiration, and breeding across all of the environments. Water temperature (WT) was critical in this process and was positively correlated with body size in high-salinity environments and negatively correlated with body size in saline environments. WT was also negatively correlated with exoskeleton or external protection and respiration, contrasting with the trends observed in both saline environments. With respect to salinity, the correlation coefficient between nitrate (NO_3_^−^) and chloride (Cl^−^) reached 1, indicating a highly positive relationship within this environment. Furthermore, the correlation coefficient between NH_4_^+^ and SAL (salinity) was 0.82, revealing the significant association between these two variables. Exploring the relationships between functional traits within saline environments revealed subtle interactions. In saline environments, we observed a positive correlation between the WT and traits such as respiration, morphology, swimming ability, and thermal preference, while other traits displayed negative correlations. However, in high-salinity environments, the relationships between the WT and biological traits were positively correlated with pH, adding complexity to the intricate ecological relationships (Figure 4c).

For the benthic functional traits in the three different habitats, we conducted CCA and selected the first two principal components. Distinct community divisions of benthic organisms were observed in environments with different salinities. The first two principal components explained 48.08% and 27.05% of the total data variance, respectively, for a total of 75.13%. The variable correlation plot revealed a strong positive correlation between PC1 and the variables WT and pH, while there was a negative correlation with NH_4_^+^. NH_4_^+^ mostly influenced Rhynchobdellida and Hemiptera. PC2 exhibited a positive correlation with TDS and SAL and a negative correlation with TP. The distributions of Ephemeroptera and Diptera were primarily affected by TP (Figure 5a,b).

### 3.3. Correlations and Comparisons of Species and Functional Diversity

In our study, we conducted a spatial analysis of the species diversity and functional diversity of benthic animals in three lakes with different salinity characteristics. The results showed that four of the eight diversity indices exhibited significant differences across the different salinity environments. These indices included the functional evenness index (FEve), functional dispersion index (FDiv), and Pielou’s evenness index (J). Notably, FEve, FDiv, and J did not differ significantly among the different salinity environments.

For species diversity, the Pielou evenness index (J) reached its highest value in freshwater environments. Regarding functional diversity, the results indicated that FDiv and the functional richness index (FRic) reached their highest levels in brackish water environments. Conversely, the functional dispersion index (FDis) and Rao’s Q peaked in high-salinity environments. FEve was highest in freshwater environments, but lowest in brackish water environments. Additionally, we observed significant intragroup differences for each index, with FDis and Rao Q showing highly significant differences between high-salinity and freshwater environments and significant differences from saline environments (Figure 6).

## 4. Discussion

### 4.1. Mechanisms of the Salinity Stress Response in Benthic Organisms

We explored the benthic aquatic organism community composition and functional traits in three salinity gradients in the Altai region. The results showed that the benthic organisms in freshwater environments exhibited greater species diversity and functional diversity than those in saltwater environments. In contrast, increasing salinity accelerates community changes among benthic organisms, reducing species diversity and decreasing functional homogenization.

High-salinity environments typically constrain the diversity of benthic organisms, resulting in only a few specialized organisms capable of surviving under these conditions [33]. In our study of high-salinity environments, we observed a significant reduction in species diversity, ultimately leaving only four orders. This phenomenon is directly related to the survival challenges posed by high-salinity conditions, including increased osmotic pressure and the adverse effects of toxic salts on these organisms [34,35]. Therefore, only specific benthic organisms can thrive and possess particular ecological and physiological characteristics to adapt to high-salinity environments [36,37]. However, the impact of saline environments on benthic organism diversity may vary depending on habitat characteristics. In saline environments, two orders, Coleoptera and Diptera, accounted for a greater proportion of the total population compared with freshwater environments. This suggested that they can adapt to varying salinity conditions, establishing them as key species for benthic communities in saline environments. These findings emphasized the critical function of habitat diversity across various salinity environments.

In high-salinity and salt–alkali environments, benthic organisms employ various adaptation strategies to cope with extreme salinity, ensuring survival and reproduction under these challenging conditions. For example, some species migrate more frequently to seek lower-salinity waters, avoiding the adverse effects of high salinity [34,38,39]. In contrast, other benthic organisms adopt certain behaviors to adapt to high salinity, seek refuge to mitigate exposure to these conditions, and reduce the need to leave high-salinity environments [40,41]. The variations in the functional traits of benthic organisms in environments with different salinities imply great differences in their roles in organic matter decomposition, their position in the food web, and their contributions to ecosystem functionality [42]. These differences are essential for maintaining the stability and functionality of ecosystems. The adaptability of benthic organisms to various saline–alkali conditions is reflected in their diverse biological and functional traits, providing a degree of assurance of ecosystem stability and adaptability.

### 4.2. Influence of Abiotic Factors on Zonal Distribution Characteristics

When investigating the impact of salinity changes on benthic organisms, we found that salinity and total dissolved solids (TDS) significantly affected the functional traits of benthic organisms in different environments. Various functional traits were prevalent in freshwater environments, with some accounting for more than 50% of the relative abundance. However, in higher salinity environments, the relative proportions of each functional trait gradually decreased, reflecting the adaptability of benthic organisms to saline conditions [43]. In saline environments, the proportion of “light calcification” traits associated with the exoskeleton or external protection was significantly greater than that associated with other traits, suggesting the vital role of “light calcification” traits in the survival and reproduction of benthic organisms [44]. Specific functional traits related to adaptations such as swimming, small body size (<9 mm), and respiratory structures such as respiratory tubes and dorsal plates dominated under high-salinity conditions. These traits represent essential strategies for thriving under high-salinity conditions. In highly saline and salt–alkali environments, benthic organisms undergo morphological changes. For example, some species may develop thicker exoskeletons or external protection mechanisms in high-salinity environments to reduce water evaporation and prevent salt intrusion [45,46]. Concurrently, other species may reduce body size to minimize water loss and enhance adaptability to elevated salinity conditions [14,47,48]. Research indicates that benthic organisms exhibit physiological adaptations in response to high-salinity environments. These adaptive strategies involve increased salt tolerance, alterations in metabolic pathways to cope with elevated salinity, and mechanisms to reduce water loss [49]. These physiological adjustments are crucial for maintaining water and ion balance within high-salinity environments.

Our results also revealed a significant correlation between ammonia ions and the functional traits of benthic organisms in environments with different salinities. This correlation exhibited strikingly different trends between freshwater and high-salinity environments. In freshwater environments, ammonia ions exhibit a significant positive correlation with functional traits such as exoskeletal or external protection, habitat preference, morphology, and feeding habits. This positive correlation aligns well with the characteristics of freshwater environments, where ammonia ions likely act as a critical nitrogen source, influencing the ecological niche allocation and functional traits of benthic organisms [50,51]. Benthic organisms may adjust the level of their exoskeleton or external protection based on ammonia ion abundance to better adapt to ammonia-dominated living conditions [52].

Moreover, ammonia ion concentrations may influence benthic organism habitat selection, morphology, and feeding habits to optimize resource utilization and survival strategies [53]. However, as we transition to high-salinity environments, ammonia ion concentrations negatively correlate with functional traits such as body size, respiration, and reproduction. This implies that under high-salinity conditions, ammonia ions may no longer be the primary ecological resource or that their concentration may detrimentally affect the survival and reproduction of benthic organisms [54,55]. As a result, benthic organisms may adopt different strategies, such as reducing body size to minimize water and energy consumption or altering reproductive methods to adapt to these extreme conditions.

Water temperature is crucial for determining the biological characteristics of benthic organisms in environments with various salinities [56]. The negative correlation between water temperature and specific functional traits, such as exoskeletal or external protection and respiratory structures, suggests that high temperatures may adversely affect these functional traits, particularly in high-salinity environments. This may be due to the increased need for better temperature adaptation and water retention mechanisms under high-temperature conditions, resulting in a reduction in exoskeleton or external protection traits to minimize heat accumulation [57,58]. Moreover, the influence of water temperature on dissolved oxygen levels should be considered. Changes in respiratory modes may be geared towards adapting to high-temperature environments by enhancing oxygen uptake or reducing water loss. These findings highlight the complexity of adaptive strategies for benthic organisms in environments with different salinities, where these strategies are intricately linked to alterations in functional traits and adjustments in ecological roles and relationships.

### 4.3. Relationship between Biodiversity and Functional Diversity

In this study, we explored the species diversity and functional trait diversity of benthic organisms in environments with different salinities and uncovered various patterns and relationships. First, we observed significant differences in four diversity indices across different salinity environments. These differences underscore the variations in benthic organism diversity and functional traits across environments with different salinities. The benthic organisms exhibited greater ecological niche differentiation in hypersaline environments than in freshwater environments, suggesting that they adopted a specific adaptive strategy to cope with extreme salinity conditions. This adaptive strategy is reflected in several indices of functional diversity, including higher functional richness, functional dispersion, functional separation, and Rao’s Q. The hypersaline environment may have exerted specific pressures that led to more functional differences within the community of organisms, prompting ecotope differentiation and providing the impetus for functional diversity throughout the community [59,60]. This ecological niche differentiation could be an attempt to avoid competition, maximize resource utilization in high-salt environments, and enhance the overall stability of ecosystems [61,62].

In contrast, benthic organisms exhibit moderate levels of ecotope differentiation in brackish water environments, indicating some adaptation to relatively high salinity conditions. Communities of organisms in such environments may be more focused on vertical adaptation, i.e., forming adaptations within species, rather than horizontal ecological niche differentiation between species. The relatively small functional differences may reflect the specialization of adaptive traits to cope with relatively stable salinity conditions more efficiently [63,64].

Furthermore, the functional diversity of benthic organisms was lowest in freshwater environments. This difference may be attributed to the lower environmental stress in freshwater environments than in high-salinity environments, which results in greater functional similarity between benthic organisms. This relatively low functional diversity may imply less ecological niche differentiation, which has implications for the functional stability of the ecosystem as a whole [65,66]. In such environments, benthic organisms may emphasize horizontal adaptation, i.e., the coutilization of resources than differentiation at specific ecological niches [67].

## 5. Conclusions

In this study, we investigated the impact of saline–alkaline environments on benthic organism diversity and functional diversity. This research involved the analyses of biological characteristics, functional traits, and ecological relationships in various salinity environments, providing valuable insights for a comprehensive understanding of these ecosystems.

First, we observed significant differences in the biological characteristics of benthic organisms in environments with different salinities. In saline lake environments, the increased prevalence of traits related to mild hardiness indicated adaptive responses to higher salinity conditions. Furthermore, in high-salinity environments, we identified the widespread presence of specific functional traits, including swimming ability, decreased body size, and adaptability to air breathing. These adaptive strategies enhance the survival capacity of organisms under extreme conditions.

Second, the adaptation and diversity of benthic organisms in saline–alkaline environments hold ecological significance. Changes in biological characteristics and functional traits across environments with different salinities imply distinct roles in the ecological functioning and balance of ecosystems. Certain benthic organisms employ unique adaptation strategies in high-salinity environments, which mainly contribute to maintaining the ecological balance under extreme conditions.

Our study provides a new theoretical and empirical framework for a deeper understanding of ecosystems in saline–alkaline environments. The identified adaptations of organisms, such as enhanced swimming ability and adaptability to air breathing at high salinity, offer practical insights for managing the ecological consequences of climate change. Recognizing the distinct roles of benthic organisms across different salinity levels is crucial for enhancing ecosystem resilience. This research establishes a practical foundation for conservation and management, aiding global efforts to tackle the challenges posed by environmental changes. Ongoing studies on benthic organisms and functional diversity in saline–alkaline environments will advance our knowledge of effective preservation strategies. We encourage future research to explore benthic organisms and functional diversity in saline–alkaline environments to better understand these critical ecosystems.

## Figures and Tables

**Figure 1 biology-13-00027-f001:**
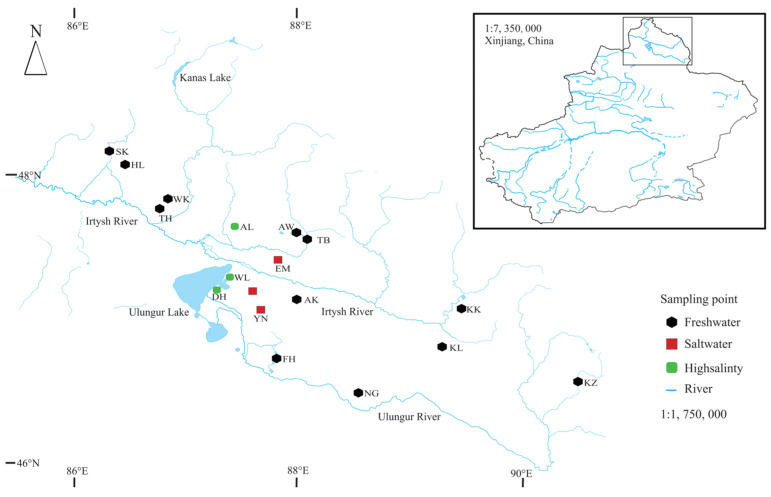
Map of the study site showing the location of the sampling site. Black hexagons indicate freshwater salinities of 0–5; red squares indicate brackish water with salinities of 5–20; and green rounded rectangles indicate hypersaline environments with salinities > 20.

**Figure 2 biology-13-00027-f002:**
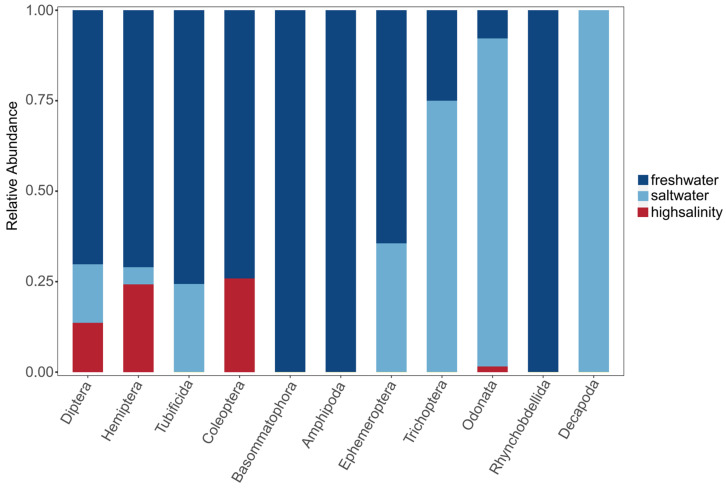
Cumulative percentage of each order of benthic organisms in environments with different salinities.

**Figure 3 biology-13-00027-f003:**
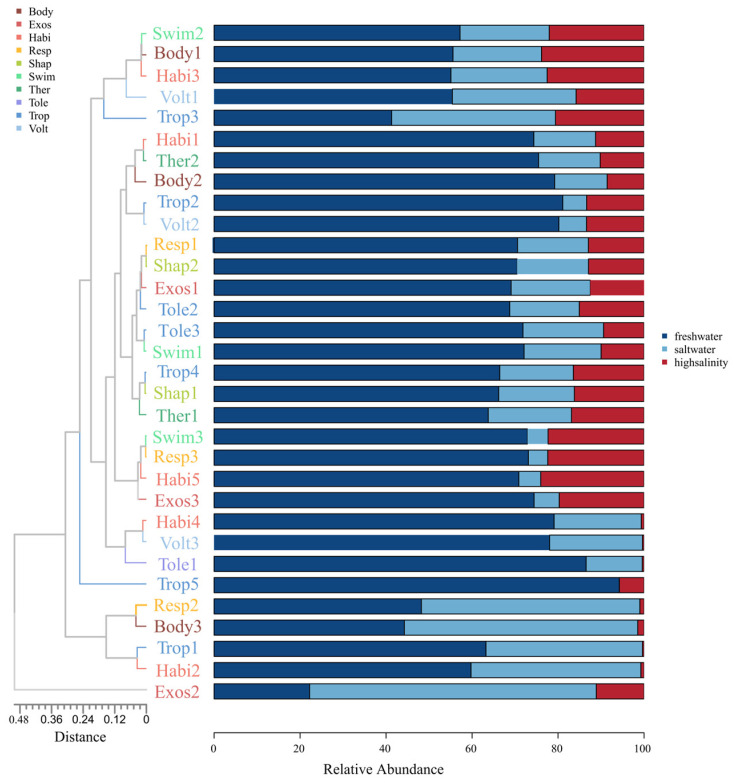
Distribution and clustering of functional trait percentages of benthic organisms in environments with different salinities.

**Figure 4 biology-13-00027-f004:**
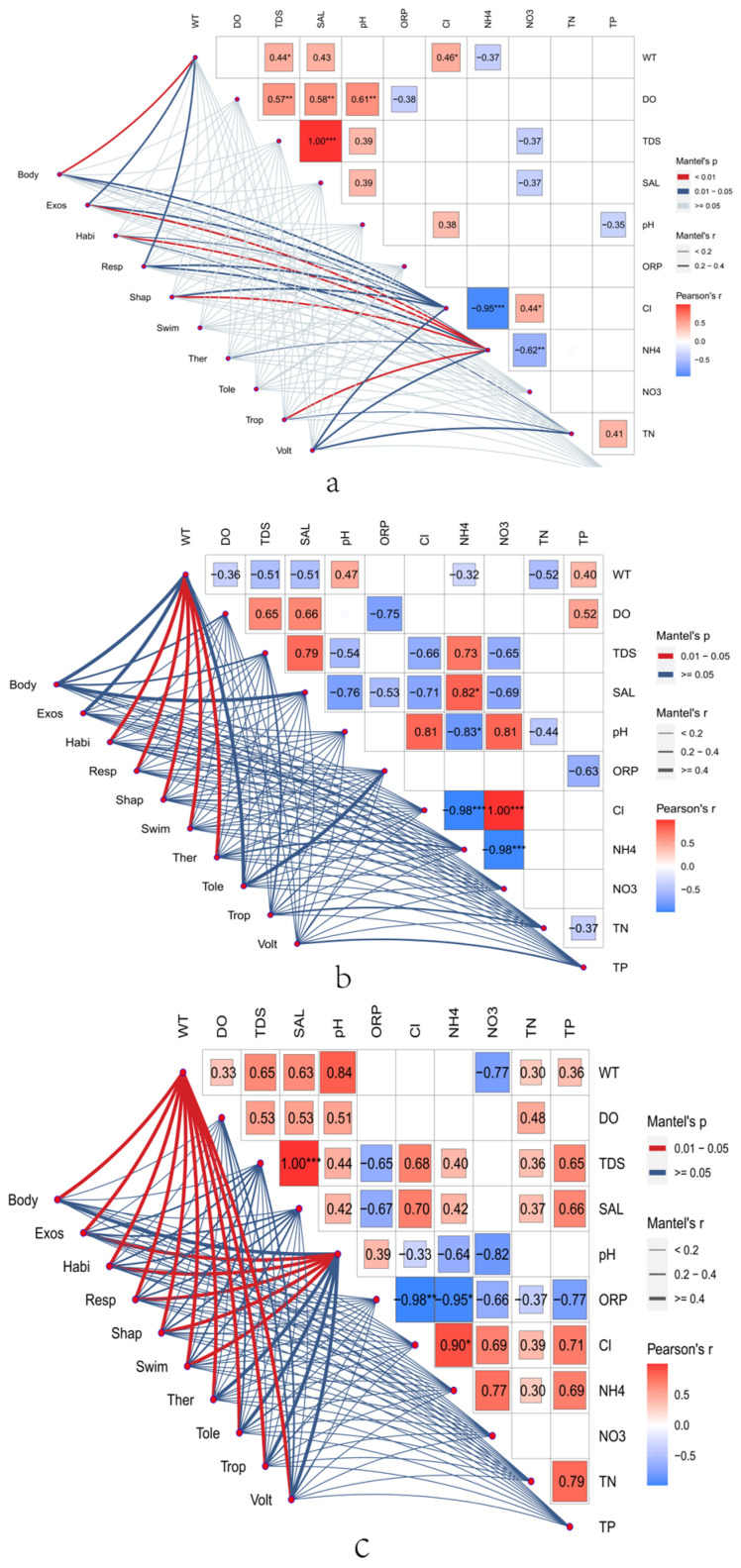
In the Mantel test, (**a**) represents freshwater environments, (**b**) represents brackish environments, and (**c**) represents hypersaline environments; *p*-values were adjusted using Benjamini–Hochberg false discovery correction; * Benjamini–Hochberg-adjusted 0.01 ≤ *p* < 0.05; ** Benjamini–Hochberg-adjusted 0.001 ≤ *p* < 0.01; *** Benjamini–Hochberg-adjusted *p* < 0.001.

**Figure 5 biology-13-00027-f005:**
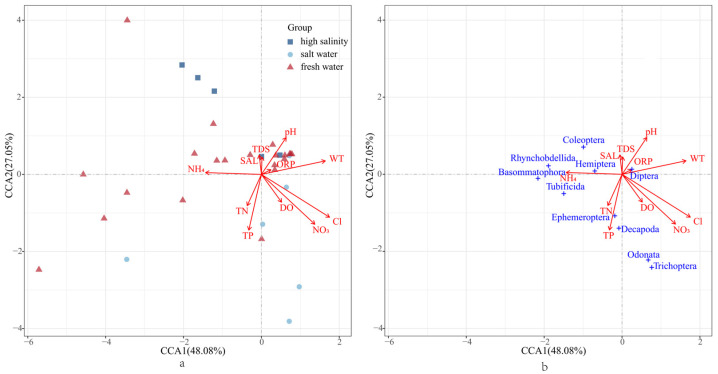
CCA analysis plot of benthic organism distributions in environments with different salinities: (**a**) sampling site information; (**b**) biotope information.

**Figure 6 biology-13-00027-f006:**
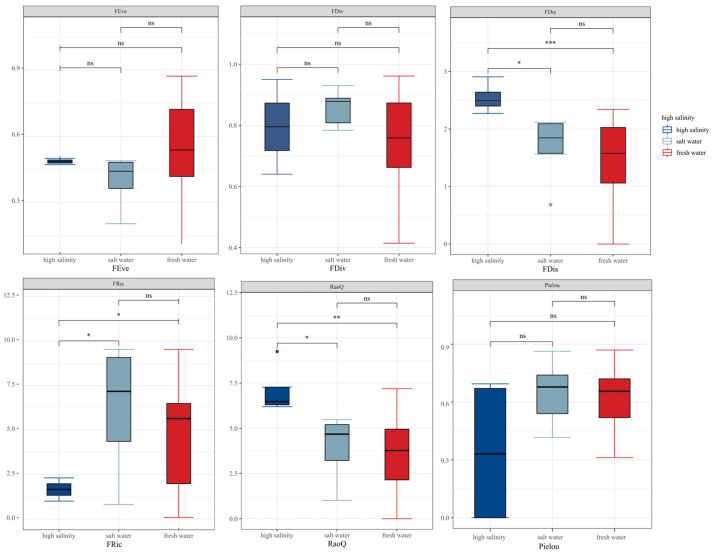
Benthic biodiversity indices and functional diversity indices in environments with different salinities; ns denote *p* > 0.05; * denote *p* ≤ 0.05; ** denote *p* ≤ 0.01; *** denote *p* ≤ 0.001.

**Table 1 biology-13-00027-t001:** Delineation criteria for the classification of benthic organisms via functional characteristics.

Trait	Trait State	Code	Score
Voltinism	Semivoltine	Volt1	1
	Univoltine	Volt2	2
	Bior multivoltine	Volt3	3
Body size	Small (<9 mm)	Body1	1
	Medium (9–16 mm)	Body2	2
	Large (>16 mm)	Body3	3
Exoskeleton or external protection	Soft-bodied forms	Exos1	1
	Lightly sclerotized	Exos2	2
	Heavily sclerotized	Exos3	3
Shape	Streamlined	Shap1	1
	Not streamlined	Shap2	2
Respiration	Tegument	Resp1	1
	Gills	Resp2	2
	Air (spiracles, tracheae, plastrons)	Resp3	3
Swimming ability	None	Swim1	1
	Weak	Swim2	2
	Strong	Swim3	3
Thermal preference	Cool eurythermal	Ther1	1
	Cool/warm eurythermal	Ther2	2
Habit	Burrower	Habi1	1
	Climber	Habi2	2
	Sprawler	Habi3	3
	Clinger	Habi4	4
	Swimmer	Habi5	5
Trophic habit	Collector-gatherer	Trop1	1
	Collector-filterer	Trop2	2
	Scraper	Trop3	3
	Predator	Trop4	4
	Shredder	Trop5	5
Pollution resistance value	Weak	Tole1	1
	Moderate	Tole2	2
	Strong	Tole3	3

## Data Availability

The data supporting this study’s findings are available from the corresponding authors upon reasonable request.

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
