# Peer review of "Ecology of Saline Watersheds: An Investigation of the Functional Communities and Drivers of Benthic Fauna in Typical Water Bodies of the Irtysh River Basin"

_biology, 2024, doi:10.3390/biology13010027_

Round 1
Reviewer 1 Report
Comments and Suggestions for Authors
The article is devoted to the biodiversity and functional structure of macrozoobenthos assemblages from typical water bodies in the Altai Region. The article is interesting and dedicated to current topics and contributes to a deeper understanding of ecosystems in saline-alkaline environments. This information is important for the conservation and management of these ecosystems and offers substantial support in keeping with global environmental changes.
The text is written in a clear and readable English, as I can assess. Statistical analysis is detailed and adequate to the aims of the authors. However, there are a number of comments regarding statistical processing, that will be presented below.
It is necessary make some corrections in the manuscript:
Materials and Methods
1. The exact number of samples that are analyzed in this study absent in the text. This shold be improved.
2. What is the macrozoobenthos sample area in m²? There were a very strange samples with volume of 0.3 m³. Explain how these unusual samples were taken? Is it really to a depth of 0.3 m inside the sediments?
3. Figure 1 is not of good enough quality. It is very small and there are no geographical names labeled on it. At least rivers and big lakes should be signed.
4. The authors calculated a lot (8!) of species diversity indexes. The number of indexes is really excessive. It is not clear what biological meaning their calculation makes? Stop please on two contrasting, in terms of data analysis, indexes - Shannon and Simpson. This will be quite sufficient for the presented research. Calculation of a large number of similar indexes does not provide useful information for explaining the structure of the studied assemblages.
Results
1. Captions to figures 2, 3 and 5 are not informative, it should be made more detailed.
2. Links to figures 2, 4, 5, 6, 7 are absent in the text of the article.
3. Figure 4 should be deleted. It is extremely difficult to understand and therefore low in information. Essentially, this is the primary analysis of data, from which only most significant and reliable values should be selected and presented in a table. In this form, this Figure can't be left in the article.
4. Figure 5 will be more informative with dividing into (A) and (B). Sampling stations should be represented on (A), and taxonomic groups should be represented on (B). Mixing sampling stations and taxonomic groups on one figure is not the best way to present these data.
5. In section 3.3. "Correlations and Comparisons of Species and Functional Diversity" should be left to compare only two species diversity indexes: Shannon and Simpson. Calculation and comparison of a large number of similar indexes does not provide useful information for description of structure of studied assemblages.
Figure 7 about the correlation between large number of species diversity indices haven't biological sense. This could be presented in a methodological work on the selection of indices for monitoring only.
Discussion
This part of the manuscript should be slightly modified in accordance with the changes in the Results.
I recommend this manuscript for publication in “Biology” after major revision.
Author Response
Dear Editors and Reviewers:
We greatly appreciate your professional review of our article. As you mentioned, there are several problems that need to be addressed. According to your suggestions, we have made extensive corrections to our previous draft. We tried our best to improve the manuscript and made some changes to it. These changes will not influence the content or framework of the paper. Here, we did not list the changes but marked them in red in the revised paper. We appreciate the sincerity of the Editors and Reviewers and hope that the corrections will be met with approval.
If there are any other modifications we could make, we would like very much to modify them, and we truly appreciate your help.
The reviewer comments are listed below in italicized font, the responses are in normal font, and changes/additions to the manuscript are given in red text.
Reviewer #1: The exact number of samples that are analyzed in this study absent in the text. This shold be improved.
- Response to comment:We apologize for not including the sample size in the Materials and Methods section. In the latest revision, we have added a description of these data.
Reviewer #2: What is the macrozoobenthos sample area in m²? There were very strange samples with a volume of 0.3 m³. Explain how these unusual samples were taken? Is it really to a depth of 0.3 m inside the sediments?
- 2.Response to comment:The correct measurement for the sample area is 0.3 m2. The unusual samples with a reported volume of 0.3 m³ were indeed an error, and we apologize for any confusion. The correct information is that the sample area is 0.3 m2, and there was no sampling to a depth of 0.3 meters inside the sediments. We have corrected this error in the manuscript.
Reviewer #3: Figure 1 is not of good enough quality. It is very small, and there are no geographical names labeled on it. At least rivers and big lakes should be signed.
- 3.Response to comment:We have changed Figure 1 to include the names of the major rivers and lakes in the watershed and increased the image content and labeling size.
Reviewer #4: The authors calculated a lot (8!) of species diversity indexes. The number of indexes is really excessive. It is not clear what biological meaning their calculation makes? Stop please on two contrasting, in terms of data analysis, indexes - Shannon and Simpson. This will be quite sufficient for the presented research. Calculation of a large number of similar indexes does not provide useful information for explaining the structure of the studied assemblages.
- 4.Response to comment:In response to your suggestion, we have revised the manuscript accordingly, eliminating excessive indices. This adjustment not only aligns with your recommendation but also enhances the clarity and interpretability of our results.
Reviewer #5: Captions to figures 2, 3 and 5 are not informative, it should be made more detailed.
- 5. Response to comment:We apologize for the oversight of incomplete information in the image captions. We have redescribed the caption information for Figures 2, 3, and 5 with more detail and accuracy.
Reviewer #6: Links to figures 2, 4, 5, 6, 7 are absent in the text of the article.
- 6. Response to comment:An oversight caused us to forget to add the image link. We apologize for this. This information has beento the image in the main text.
Reviewer #7: Figure 4 should be deleted. It is extremely difficult to understand and therefore low in information. Essentially, this is the primary analysis of data, from which only most significant and reliable values should be selected and presented in a table. In this form, this Figure can't be left in the article.
- 7.Response to comment:We appreciate your careful consideration of our manuscript. Regarding Figure 4, which presents the results of the Mantel test analysis, we understand your concern about its complexity. However, this figure is crucial in illustrating the relationships within our data, providing valuable insights into the core analysis.
The complexity in Figure 4 arises from the intricate nature of the relationships explored through the Mantel test. Deleting the figure would remove a significant visual representation of our primary analysis. To address the concern about clarity, we are committed to revising the Constitution to enhance its readability without sacrificing the essential information it conveys. We will simplify the presentation and ensure that the key findings are more accessible to the readers.
Reviewer #8: Figure 5 will be more informative with dividing into (A) and (B). Sampling stations should be represented on (A), and taxonomic groups should be represented on (B). Mixing sampling stations and taxonomic groups on one figure is not the best way to present these data.
- 8.Response to comment:We appreciate your feedback and agree that dividing the figure into two parts, (A) representing sampling stations and (B) representing taxonomic groups, will enhance the clarity and informativeness of the figure. We have revised Figure 5 accordingly, with (A) now displaying the distribution of sampling stations and (B) illustrating the taxonomic groups. This modification will significantly improve the visual presentation and facilitate a clearer understanding of the data.
Reviewer #9: In section 3.3. "Correlations and Comparisons of Species and Functional Diversity" should be left to compare only two species diversity indexes: Shannon and Simpson. Calculation and comparison of a large number of similar indexes does not provide useful information for description of structure of studied assemblages.
- 9. Response to comment:We appreciate your guidance in refining the focus of our analysis. In response to your suggestion, we have removed the calculation and comparison of multiple species diversity indices. This revision enables a more straightforward presentation of the critical diversity metrics, contributing to a more focused and informative discussion.
Reviewer #10: Figure 7 about the correlation between large number of species diversity indices haven't biological sense. This could be presented in a methodological work on the selection of indices for monitoring only.
- 10. Response to comment:Based on your suggestion, we have removed Figure 7 and the corresponding description from the manuscript. We acknowledge that the correlation between many species diversity indices may not have direct biological significance.
Reviewer #11: This part of the manuscript should be slightly modified in accordance with the changes in the Results.
- 1Response to comment:We appreciate your reminder to align the manuscript with the changes in the Results section. Based on your suggestions, we have carefully revised the relevant parts of the manuscript to ensure consistency with the updated results.
We tried our best to improve the manuscript by making changes marked in red in the revised paper that did not influence the content or framework of the paper. We greatly appreciate the work of the Editors and Reviewers, and we hope that the corrections will be met with approval. Once again, thank you very much for your comments and suggestions.
Reviewer 2 Report
Comments and Suggestions for Authors
A very interesting paper based on a large dataset. The English and the construction are good and I enjoyed reading it, but there are a few changes I would suggest that might make it a little more understandable and perhaps of more interest to a wider readership.
Some points to note:
1. Abstract (and Intro) need to state what range of salinity is being investigated: otherwise mentions of ‘freshwater’ are misleading as this represents only one end of the spectrum.
2. I feel it is worth referring to the Remane curves and citing for example some of the (admittedly few) works in TFTWs. Remane’s work should at least give a model for community (S) response over salinity range, and his boundaries do give some guide as the cut-offs (see also Figs 1 and 4 and comment 3 below). It also gives a basis for comparison of open ended systems estuaries) and these, closed systems.
3. I would like to see references/citations for these traits in Table 1. Why were these chosen? Can the authors justify the boundaries/cut-offs between categories? This would also allow a wider discussion and comparison with other similar trait-based studies as to those traits which best respond to environmental cues.
4. Line 228 ‘Mild hardening traits’. What is meant by this? (and see comment 3 above). Why the quotation marks?
5. Lines 333-4: this is not really saying anything useful. What different attributes are these based on and what does this tell us about other system properties such as resilience/resistance or structure?
6. Lines 420-2 - heat regulation: by ectotherms? Reword or clarify.
On the version I received, the legends for Fig axes were so small as to be unreadable, and the highlights (in Fig 4 for example) come up in Chinese which unfortunately I cannot read.
Comments on the Quality of English LanguageGood
Author Response
Dear Editors and Reviewers:
We greatly appreciate your professional review of our article. As you mentioned, there are several problems that need to be addressed. According to your suggestions, we have made extensive corrections to our previous draft. We tried our best to improve the manuscript and made some changes to it. These changes will not influence the content or framework of the paper. Here, we did not list the changes but marked them in red in the revised paper. We appreciate the sincerity of the Editors and Reviewers and hope that the corrections will be met with approval.
If there are any other modifications we could make, we would like very much to modify them, and we truly appreciate your help.
The reviewer comments are listed below in italicized font, the responses are in normal font, and changes/additions to the manuscript are given in red text.
Reviewer #1: Abstract (and Intro) need to state what range of salinity is being investigated: otherwise mentions of ‘freshwater’ are misleading as this represents only one end of the spectrum.
- Response to comment:In response to your feedback, we have revised the Abstract and Introduction to state the range of salinities investigated explicitly. We hope that these additions will provide a clearer understanding of the scope of our study, addressing the concern about potential misinterpretation of the term 'freshwater.'
Reviewer #2: I feel it is worth referring to the Remane curves and citing for example some of the (admittedly few) works in TFTWs. Remane’s work should at least give a model for community (S) response over salinity range, and his boundaries do give some guide as the cut-offs (see also Figs 1 and 4 and comment 3 below). It also gives a basis for comparison of open ended systems estuaries) and these, closed systems.
- 2.Response to comment:We have incorporated references to Remane's work, providing a model for community response over a salinity range. This inclusion will contribute to a more comprehensive understanding of our research context and provide a foundation for discussions on community responses to varying salinity conditions.
Reviewer #3: I would like to see references/citations for these traits in Table 1. Why were these chosen? Can the authors justify the boundaries/cut-offs between categories? This would also allow a wider discussion and comparison with other similar trait-based studies as to those traits which best respond to environmental cues.
- 3.Response to comment:We consulted a large body of relevant academic literature and selected representative studies to provide a basis for these traits. These references not only help to increase the scientific rigor of our study but also enable broader discussions and comparisons with similar trait-based studies.
Reviewer #4: Line 228 ‘Mild hardening traits’. What is meant by this? (and see comment 3 above). Why the quotation marks?
- 4.Response to comment:Following your suggestion, we have revised this term to provide a more explicit description, stating " Lightly covered exoskeleton." The quotation marks have been removed to eliminate any unnecessary ambiguity.
Revewer #5: Lines 333-4: this is not really saying anything useful. What different attributes are these based on and what does this tell us about other system properties such as resilience/resistance or structure?
- 5.Response to comment:In accordance with your suggestion, we removed this section because it did not provide additional helpful information.
Reviewer #6: Lines 420-2 - heat regulation: by ectotherms? Reword or clarify.
- 6.Response to comment:We have addressed your suggestion by rephrasing "heat regulation" to "temperature adaptation" to provide more precise and accurate language. This modification enhances the precision of our description.
Reviewer #7: On the version I received, the legends for Fig axes were so small as to be unreadable, and the highlights (in Fig 4 for example) come up in Chinese which unfortunately I cannot read.
- 7.Response to comment:We appreciate your feedback and have made the necessary adjustments to ensure that the legends are now readable and that the highlights are presented in a language accessible to all readers. Thank you for your time and thorough review.
We tried our best to improve the manuscript by making changes marked in red in the revised paper that did not influence the content or framework of the paper. We greatly appreciate the work of the Editors and Reviewers, and we hope that the corrections will be met with approval. Once again, thank you very much for your comments and suggestions.
Reviewer 3 Report
Comments and Suggestions for Authors
I would like to congratulate the authors for the tremendous and thoughtful and work dedicated for the preparation of this manuscript. I found it a very interesting and insightful read. In general, the article is well written using a proper English. However, I would take a proof read of the Discussion in order to better organize the section and avoid some repetitions.
I leave you just below a small list of suggestions and comments that, could contribute to the clarity and readability of the article:
Title: I am not familiar with the use of “typical water bodies”. I would avoid the use of “typical” in the title and rephrase it to a more standard term. However, if the authors are certain of its use in this context, please ignore this comment.
L22: “were higher”
Key-words: I would avoid the use of terms already included in the title such as Saline or Benthic fauna. I suggest to look for synonyms that may increase the findability of the article in search-engines.
Figure 1: I would recommend to the authors to include a location maps and/or a coordinate grid to better locate the area of study for the readers that are not familiar with the area of study.
L113: when using e.g., to enumerate it is not followed by etc.
L114: I would shift engender to constitute.
L117: rephrase “based on credible information” and, if possible, include the list of articles/dichotomic keys and/or books used for the identification process.
L122: avoid repetition of “encompassing”.
L135: “Table 1. Binary biological trait variables and categories of macroinvertebrate communities.” May be a typo from previous versions of the manuscript?
In addition in the Trophic habit trait, why “Scrapers” are the only ones in plural?
Moreover “Tolerance value” is a trait that make reference to which kind of tolerance? Please specify somehow.
L141: benthic organisms
L202: why linkET package is the only one without “”?
Figure 2: Please increase the size of the font in the figure.
Figure 4 & 5 are not cited in the text. Moreover, I would put all 3 subfigures in vertical and increase their size to make them more readable.
L306: from now on the term “salinity environments” is often present in the text. I am not quite familiar with it, unless the authors are completely sure I would change it for other term such as salinity conditions, hyper-saline environments or as used later in this section high-salinity environments.
Figure 6: not cited in the text. Please increase the size of the font in this figure.
Figure 7: not cited in the text.
L343: what do you mean by “more excellent” species?
L347: be careful with the use of populations. Maybe here is more appropriate the use of specialized organisms?
L352: again here is used populations. Unless the authors are talking about different populations of the same species that present different adaptations to hypersaline environments in comparison of other populations of the same species, I would reformulate to communities or range of species.
L355-356: The phrase “In high-salinity environments, we noted that two orders, Trichoptera and Odonata, dominated under different salinity gradients” it is not clear. Please better explain.
L357: The phrase “establishing them as key species in benthic communities.” is too broad, benthic communities where?
L385-393: this paragraph, although it contains all the necessary references and some interesting ideas, I suggest to better connect the different statements to increase readability.
L415-426: in this paragraph, talking about compromise between temperature and external protection and respiratory systems, I am missing to include the fact that temperature may also affect oxygen solubility and all its possible implications in the adaptations presented.
L458: again, the term “salinity environments”.
L476: Please specify how the results/knowledge obtained by this study may offer a substantial support in addressing global environmental changes.
Comments on the Quality of English LanguageAs previously said, in general, the article is well written using a proper English. However, I would try to improve the Discussion section to better organize the presented ideas and avoid some repetitions.
Author Response
Dear Editors and Reviewers:
We greatly appreciate your professional review of our article. As you mentioned, there are several problems that need to be addressed. According to your suggestions, we have made extensive corrections to our previous draft. We tried our best to improve the manuscript and made some changes to it. These changes will not influence the content or framework of the paper. Here, we did not list the changes but marked them in red in the revised paper. We appreciate the sincerity of the Editors and Reviewers and hope that the corrections will be met with approval.
If there are any other modifications we could make, we would like very much to modify them, and we truly appreciate your help.
The reviewer comments are listed below in italicized font, the responses are in normal font, and changes/additions to the manuscript are given in red text.
Reviewer #1: Title: I am not familiar with the use of “typical water bodies”. I would avoid the use of “typical” in the title and rephrase it to a more standard term. However, if the authors are certain of its use in this context, please ignore this comment.
- Response to comment:We have carefully considered your suggestion to avoid using "typical" in the title. After thorough deliberation, we have maintained the original title, "Ecology of Saline Watersheds: An Investigation of the Functional Communities and Drivers of Benthic Fauna in Typical Water Bodies of the Irtysh River Basin." We understand the importance of clarity and precision in the title, and we believe that the term "typical" accurately reflects the representative nature of the water bodies under study.
Reviewer #2: L22: “were higher”
- 2.Response to comment: We have noted language error you described, and we appreciate your guidance. We have since addressed this issue by making corrections to the manuscript.
Reviewer #3: Key-words: I would avoid the use of terms already included in the title such as Saline or Benthic fauna. I suggest to look for synonyms that may increase the findability of the article in search-engines.
- 3.Response to comment:We appreciate your suggestion to avoid terms already in the title, such as "Saline" and "Benthic fauna." Based on your recommendation, we have revised the keywords to enhance the article's discoverability in search engines. The updated keywords now include more specific terms related to our study, such as "Inland water bodies" and "Aquatic fauna drivers." These changes will contribute to a more comprehensive representation of our research.
Reviewer #4: Figure 1: I would recommend to the authors to include a location maps and/or a coordinate grid to better locate the area of study for the readers that are not familiar with the area of study.
- 4.Response to comment:We appreciate your input, and in response to your recommendation, we have included a coordinate grid to better assist readers in locating the study area.
Reviewer #5: L113: when using e.g., to enumerate it is not followed by etc.
- 5.Response to comment:We have duly noted your suggestion and, according to your recommendation, removed "etc." after "e.g.," to align with proper enumeration practices.
Reviewer #6: L114: I would shift engender to constitute.
- 6.Response to comment:Following your recommendation, we have made the necessary adjustment, replacing "engender" with "constitute".
Reviewer #7: L117: rephrase “based on credible information” and, if possible, include the list of articles/dichotomic keys and/or books used for the identification process.
- 7.Response to comment:We have rephrased the word and added the appropriate research paper.
Reviewer #8: L122: avoid repetition of “encompassing”.
- 8.Response to comment:We appreciate your comment and, in response to your advice, have revised the text to eliminate the redundancy of this term.
Reviewer #9: L135: “Table 1. Binary biological trait variables and categories of macroinvertebrate communities.” May be a typo from previous versions of the manuscript?
- 9.Response to comment:Binary biological trait variables and categories of macroinvertebrate communities." We appreciate your careful review, and the error has been rectified in the latest version of the manuscript.
Reviewer #10: In addition in the Trophic habit trait, why “Scrapers” are the only ones in plural?Moreover “Tolerance value” is a trait that make reference to which kind of tolerance? Please specify somehow.
- 10.Response to comment:In response to your feedback, we have revised "Scrapers" to "Scraper" to maintain consistency. Additionally, we have clarified the "tolerance value" trait by specifying it as the "pollution resistance value."
Reviewer #11: L141: benthic organisms
- 1Response to comment:We have made the necessary modifications according to your suggestion.
Reviewer #12: L202: why linkET package is the only one without “”?
- 12.Response to comment:In response to your suggestion, we have added quotations about the "linkET" package to maintain uniformity with the other packages.
Reviewer #13: Figure 2: Please increase the size of the font in the figure.
- 13.Response to comment:In response to your recommendation, we have increased the font size in the figure to enhance readability.
Reviewer #14: Figure 4 & 5 are not cited in the text. Moreover, I would put all 3 subfigures in vertical and increase their size to make them more readable.
- 14.Response to comment:We have incorporated citations in the text as per your suggestion. Additionally, we have restructured both figures by placing all three subfigures in a vertical arrangement to enhance readability. The sizes of the figures has also been increased for better visibility.
Reviewer #15: L306: from now on the term “salinity environments” is often present in the text. I am not quite familiar with it, unless the authors are completely sure I would change it for other term such as salinity conditions, hyper-saline environments or as used later in this section high-salinity environments.
- 15.Response to comment:We have carefully reviewed and adjusted the manuscript to use more accurate and specific terms throughout the text.
Reviewer #16: Figure 6: not cited in the text. Please increase the size of the font in this figure.
- 16.Response to comment:We appreciate your valuable feedback and have made the necessary adjustments by incorporating the citation and enhancing the font size in Figure 6.
Reviewer #17: Figure 7: not cited in the text.
- 17.Response to comment:We have deleted this part because Figure 7 does not convey the actual significance based on experimental and data analysis considerations.
Reviewer #18: L343: what do you mean by “more excellent” species?
- 18.Response to comment:We have duly considered your suggestion and made the appropriate modification by replacing it with the following expression: "High-salinity environments typically constrain the diversity of benthic organisms, resulting in only a few specialized organisms capable of surviving under these conditions."
Reviewer #19: L347: be careful with the use of populations. Maybe here is more appropriate the use of specialized organisms?
- 19.Response to comment:We have incorporated your advice and replaced it with "specialized organisms."
Reviewer #20: L352: again here is used populations. Unless the authors are talking about different populations of the same species that present different adaptations to hypersaline environments in comparison of other populations of the same species, I would reformulate to communities or range of species.
- 20.Response to comment:We appreciate your suggestion and have corrected this text by replacing it with "communities" to convey the intended meaning better.
Reviewer #21: L355-356: The phrase “In high-salinity environments, we noted that two orders, Trichoptera and Odonata, dominated under different salinity gradients” it is not clear. Please better explain.
- 2Response to comment:We have revised the text to address your concern by changing it to "In saline environments, we note that two orders, Coleoptera and Diptera, occupy a larger proportion relative to freshwater environments."
Reviewer #22: L357: The phrase “establishing them as key species in benthic communities.” is too broad, benthic communities where?
- 22.Response to comment:We have revised the text to increase the specificity of the manuscript, and the text now reads "establishing them as key species for benthic communities in saline environments."
Reviewer #23: L385-393: this paragraph, although it contains all the necessary references and some interesting ideas, I suggest to better connect the different statements to increase readability.
- 23.Response to comment:We have carefully addressed your suggestion and made revisions to enhance the coherence and readability of the text.
Reviewer #24: L415-426: in this paragraph, talking about compromise between temperature and external protection and respiratory systems, I am missing to include the fact that temperature may also affect oxygen solubility and all its possible implications in the adaptations presented.
- 24.Response to comment:Thank you for pointing out the need to consider the impact of temperature on oxygen solubility and its potential implications in the adaptations presented in lines 415-426. We have revised the paragraph to include this important aspect.
Reviewer #25: L458: again, the term “salinity environments”.
- 25.Response to comment:We have considered your feedback and made the necessary adjustments to improve the clarity of the text.
Reviewer #26: L476: Please specify how the results/knowledge obtained by this study may offer a substantial support in addressing global environmental changes.
- 26.Response to comment:We have incorporated a more detailed explanation in the manuscript, emphasizing that the identified adaptations of organisms, such as enhanced swimming ability and adaptability to air breathing at high salinity, provide practical insights for managing ecological consequences induced by climate change. Ongoing studies on benthic organisms and functional diversity in saline-alkaline environments will further advance our knowledge of effective preservation strategies.
We tried our best to improve the manuscript by making changes marked in red in the revised paper that did not influence the content or framework of the paper. We greatly appreciate the work of the Editors and Reviewers, and we hope that the corrections will be met with approval. Once again, thank you very much for your comments and suggestions.
Round 2
Reviewer 1 Report
Comments and Suggestions for Authors
Now main part of my comments is corrected.
Therefore I recommend this manuscript for publication in “Biology”!